Effect of low-dose radiation pre-irradiation on postoperative local chest wall recurrence of breast cancer—A retrospective study

Zeng Ruifang 1
Wang Hanyu 2
Cai Xiaojun 1
Lin Junhao 1
Li Pengfei 1
Zeng Shan 3 zengshan@sysucc.org.cn
Li Aimin 1 liaimin2005@163.com
1 Department of Oncology, Cancer Center, Southern Medical University Hospital of Integrated Traditional Chinese and Western Medicine, Southern Medical University , Guangzhou , China
2 Department of Radiation Oncology, State Key Laboratory of Oncology in South China, Collaborative Innovation, Sun Yat-sen University Cancer Center , Guangzhou , China
3 Department of Pharmacy, State Key Laboratory of Oncology in South China, Collaborative Innovation, Sun Yat-sen University Cancer Center , Guangzhou , China
Wang Jincheng
Electronic publication date: 2025 Jan 2
Publication date: 2025
Volume: 13
Electronic Location ID: e18717
Received 2024 Oct 18; Accepted 2024 Nov 25
Copyright: © 2025 Zeng et al.
Copyright year: 2025
Copyright holder: Zeng et al.
License: This is an open access article distributed under the terms of the Creative Commons Attribution License, which permits unrestricted use, distribution, reproduction and adaptation in any medium and for any purpose provided that it is properly attributed. For attribution, the original author(s), title, publication source (PeerJ) and either DOI or URL of the article must be cited.
License URL: https://creativecommons.org/licenses/by/4.0/

Keywords: Low-dose radiation, Breast cancer, Recurrence, Side effects, Survival

Funding: Traditional Chinese Medicine Bureau of Guangdong Province 20232077 Guangdong Medical Science and Technology Research Fund A2024755 This research was supported by the Traditional Chinese Medicine Bureau of Guangdong Province (project number: 20232077) and the Guangdong Medical Science and Technology Research Fund (project number: A2024755). The funders had no role in study design, data collection and analysis, decision to publish, or preparation of the manuscript.

==============================
Purpose

The purpose of this study was to determine the efficacy and safety of low-dose radiotherapy (LDR) for postoperative local chest wall recurrence of breast cancer.

Methods

The records of 52 patients with postoperative local chest wall recurrent breast cancer treated at our cancer center from January 2019 to December 2022. The t-test was used to compare the means of the LDR group and non LDR group. Categorical data were compared using the chi-square test. The Kaplan-Meier method were used to determine the factors associated the survival.

Results

Compared to patients who did not undergo LDR, patients in the LDR group showed a higher incidence of grade I side effects in their skin and soft tissue systems (p = 0.002), a significant reduction in grade II side effects (p < 0.05), and no grade III side effects. Compared with the non LDR group, the complete response rates were 42.3% vs. 38.5%, p = 0.777, the partial response rates were 53.8% vs. 50.0%, p = 0.781, and the objective relief rates were 96.2% vs. 88.5%, p = 0.833, respectively. The 3-year overall survival rate, local recurrence free survival rate, distant metastasis-free survival rate and disease-specific survival rate comparison between LDR group and non LDR group was 54.8% vs. 55.8%, p = 0.845 and 54.2% vs. 52.9%, p = 0.751, 67.9% vs. 51.9%, p = 0.097 and 39.2% vs. 49.6%, p = 0.993, respectively.

Conclusion

Compared with traditional radiotherapy, LDR pre-irradiation has better tolerance and efficacy in patients with local chest wall recurrence of breast cancer after operation.

Introduction

Breast cancer ranks first in the incidence rate of female cancer, and is also the cancer with the highest mortality rate of women in the world (Siegel et al., 2021). With the progress of treatment, the local recurrence free survival rate (LRFS) and the overall survival rate (OS) of breast cancer have improved (Siegel et al., 2021). However, about 30–40% of breast cancer patients will relapse, usually it’s be seen in HER2 overexpression or hormone dependent patients (Katsura et al., 2022).

It is reported in the literature that the annual incidence rate of local recurrence of early breast cancer after surgery is 0.6% (De Rose et al., 2022). At the same time, patients with local recurrence have the possibility of subclinical distant metastasis (Dempsey, Sandoval & Mahtani, 2023). Among patients with local recurrence of breast cancer, chest wall recurrence is the most common, accounting for 50~94% of local recurrence. It usually presents one or more asymptomatic skin or subcutaneous nodules, mainly located in surgical scars and their adjacent parts (Pedersen et al., 2022).

The treatment of breast cancer with local chest wall recurrence is one of the difficult problems in clinic, and reoperation is usually one of the first treatment options. However, some patients are large tumors and deeply invade soft tissue, so it is difficult for surgery to completely remove and it is easy to cause difficult skin anastomosis. At this time, timely local radiotherapy is usually considered. Due to the deterioration of the patient’s general condition and multiple previous treatments, it is difficult for them to tolerate conventional or high-dose radiation, especially when combined with surgery, chemotherapy, and immunotherapy. In order to reduce adverse reactions, it is necessary to limit the radiation dose, which also becomes an obstacle to the treatment effect of recurrent breast cancer.

Although radiotherapy technology has made significant progress in recent years, the application of various techniques has corresponding limitations, making it difficult to further solve the contradiction between improving tumor control and reducing normal tissue damage. Therefore, it is urgent to seek new ideas to better resolve this contradiction (Zhang et al., 2023).

With the progress of radiation protection, the biological effects of low-dose radiation (LDR) have gradually attracted attention (Dawood, Mothersill & Seymour, 2021). For a long time, people’s understanding of LDR has mainly been based on the linear no threshold hypothesis (Zielinski et al., 2009), which states that any small radiation dose can increase the risk of human carcinogenesis. However, an increasing number of studies are questioning this hypothesis, suggesting that LDR may have beneficial effects (Weissmann et al., 2022; Schubauer-Berigan et al., 2015), although the mechanism is not yet clear. Previous studies have applied LDR to head and neck tumors, non-small cell lung cancer and locally advanced breast cancer, but there is no data on chest wall recurrence of breast cancer (Gleason et al., 2013; Nardone et al., 2014; Bufi et al., 2012; Mantini et al., 2012). Whether the potential biological effects induced by LDR can be utilized to activate the protective system of normal tissues and resist radiation damage remains to be explored.

Materials and Methods

Patients

We retrospectively analyzed the clinical data of patients with postoperative local chest wall recurrent breast cancer who received LDR before radiotherapy in our Cancer Center from January 2019 to December 2022. Patients who meet the criteria will be paired using a 1:1 propensity score matching (PSM) method among patients treated at the same time, and the following variables will be used to balance patient characteristics: age, gender, histopathology differentiation, T stage, N stage, and radiotherapy and chemotherapy.

Inclusion criteria: (1) Age: 18–70 years old; (2) pathological diagnosis of invasive breast cancer; (3) patients with postoperative chest wall recurrence and it’s difficult to surgically remove after multidisciplinary consultation; (4) complete clinical data and at least one follow-up result; (5) successfully completed the treatment at our center.

Exclusion criteria: (1) Other serious diseases, such as congenital heart disease, severe pulmonary infection, type I diabetes, systemic lupus erythematosus or skin diseases that are difficult to heal; (2) the irradiated area is combined with a second primary cancer.

The study was approved by the Medical Research Ethics Committee of Southern Medical University Hospital of Integrated Traditional Chinese and Western Medicine (ID: 201902SB-016-09), and all patients signed informed consent forms for treatment and agreed to use their data for research purposes.

Treatment

According to the guidelines of the Chinese Society of Clinical Oncology (CSCO) and the National Comprehensive Cancer Network (NCCN), all patients received personalized treatment. The treatment plan includes radiation therapy (RT) and necessary systemic therapy.

RT was prescribed as a planned tumor volume (PTV) of 60–66 Gy for the total tumor volume (GTVnx); 60–68 Gy for PTV in lymph nodes (GTVnd); 50–54 Gy for PTV in the clinical target volume (CTV); 28–30 fractions. The method of target delineation and planning design of IMRT reference the International Commission on Radiation Units (ICRU) Report No. 83. The pre-irradiation dose before LDR radiotherapy is 100 mGy per fractions. Radiotherapy is performed every 6 h, once a week on Mondays and Wednesdays.

The chemotherapy regimen is recommended by a multidisciplinary collaboration group in oncology (consisting of one senior attending physician or above in radiotherapy, oncology surgery, internal medicine, imaging, and pathology) based on CSCO and NCCN guidelines, and is given after discussion. The chemotherapy regimen includes a variety of drugs (including paclitaxel, anthracycline, cyclophosphamide, carboplatin, cisplatin, capecitabine, gemcitabine, vinorelbine and fluorouracil) and anti adverse drugs based on the patient’s overall condition, as well as the reasonable selection of targeted drugs or immune drugs based on genetic testing results. The dosage and number of cycles are adjusted appropriately according to clinical experience and patient response. In this study, the main chemotherapy regimens used were AC-T (adriamycin+cyclophosphamide+paclitaxel), FAC (fluorouracil+adriamycin+cyclophosphamide), and FEC (fluorouracil+epirubicin+cyclophosphamide). The proportions of the above three regimens are 45.2%, 27.8%, and 22.9%, respectively.

Efficacy and adverse reaction evaluation

The efficacy evaluation is based on the Response Evaluation Criteria in Solid Tumors (RECIST version 1.1). Adverse reactions were evaluated using the Radiation Therapy Oncology Group (RTOG) radiation injury grading criteria.

Follow-up and clinical endpoints

Patients are followed up in outpatient or telephone interviews, with a follow-up every 3 months for the first 2 years and every 6 months after the third year. Patients are told that if they encounter any worrying situations, they should go to the hospital. The examinations include blood biochemistry tests, tumor markers, neck+chest+upper abdominal CT plain scan with enhancement, and if necessary, head MRI scan with enhancement and whole-body bone scan should be include.

The overall survival (OS) was defined as the date of initial treatment until last follow-up or death. Loco-regional recurrence-free survival (LRFS) was defined from the date of initial treatment to the date of the first chest wall locoregional recurrence, or death from any cause, or last follow-up. Distant metastasis-free survival (DMFS) was defined as the date of initial treatment until the date of first distant metastasis, or death from any cause, or last follow-up. Disease-specific survival (DSS) was defined as the date of initial treatment until death due to breast cancer, or last follow-up.

Statistical analysis

SPSS 22.0 was used as the statistical software, and t-test was used to compare the means of the two samples. Categorical data were compared using the chi-square test or Fisher’s exact probability method, with a significance marker of α = 0.05. The Kaplan–Meier method, and propensity score matching (PSM) were used to determine the factors associated the survival. p < 0.05 indicates a statistically significant difference.

Results

Patient characteristics

From January 2019 to December 2022, a total of 30 patients with postoperative local chest wall recurrent breast cancer who received LDR before radiotherapy in our cancer center were included. We excluded one patient with lung cancer at the irradiation site, one patient had no follow-up results after treatment, and two patients did not perform complete treatment due to personal reasons. A total of 26 patients were included. After 1:1 matching by PSM method, 52 patients were finally included for analysis, shown in Fig. 1. Two patients were lost to follow-up after a 3-month follow-up, with a follow-up rate of 96.2%. The baseline characteristics of the two groups of patients are shown in Table 1.

Figure 1 Flow diagram of patient selection and inclusion.

LDR, low-dose radiation; PSM, propensity score matching.

Table 1 The clinical characteristic of two group patients.

Characteristics	LDR group (n)	No LDR group (n)	p	
Median age (range)	50 (24–69)	50 (25–70)	0.982	
Gender (all female)	26	26	1	
ECOG PS			1	
0–1	18	18		
2	8	8		
Differentiation			0.905	
Luminal A	8	8		
Luminal B	8	9		
Her-2 (+++)	6	6		
Triple negative	4	3		
T stage			0.915	
1	3	3		
2	6	6		
3	8	7		
4	9	10		
N stage			1	
0	10	9		
1	7	8		
2	3	4		
3	6	5		
M stage			1	
0	26	26		
Clinical stage			0.822	
II	6	5		
III	16	17		
IV	4	4		
Chemotherapy			1	
Yes	16	16		
No	10	10		
Previous RT			1	
Yes	10	10		
No	16	16		
Note:

LDR, low-dose radiation; RT, radiation therapy.

Comparison of toxic side effects between two groups

The acute toxic reactions mainly include skin system, subcutaneous soft tissue system, blood system, and partial digestive system reactions, as shown in Table 2. The most common reactions are redness, swelling, and peeling of the skin system, congestion and swelling of the subcutaneous soft tissue system, and a decrease in white blood cells (granulocytes) in the blood system.

Table 2 The acute side effects of two group patients.

	0	I	II	III	IV	
Dermatitis						
LDR group (n)	0	18	8	0	0	
No LDR group (n)	0	7	16	3	0	
p		0.002	0.026	0.077		
Soft tissue injury						
LDR group (n)	2	21	3	0	0	
No LDR group (n)	1	10	14	1	0	
p	0.556	0.002	0.001	0.317		
Granulocytopenia						
LDR group (n)	5	12	7	2	0	
No LDR GROUP (n)	4	7	6	7	2	
p	0.717	0.150	0.510	0.070	0.153	
Anemia						
LDR group (n)	13	7	5	1	0	
No LDR group (n)	12	4	8	2	0	
p	0.781	0.308	0.337	0.552		
Thrombocytopenia						
LDR group (n)	14	10	2	0	0	
No LDR group (n)	14	5	7	0	0	
p	1	0.126	0.070			
Swallowing discomfort						
LDR group (n)	8	18	0	0	0	
No LDR group (n)	6	16	4	0	0	
p	0.532	0.846	0.039			
Nausea						
LDR group (n)	8	18	0	0	0	
No LDR group (n)	12	13	1	0	0	
p	0.358	0.158	0.317			
Note:

LDR, low-dose radiation.

Compared to patients who did not receive LDR, more patients in the LDR group exhibited only grade I side effects in their skin and soft tissue systems, which did not reach grade II, a significant reduction in grade II side effects, and no grade III side effects.

In addition, there were four cases of grade II esophagitis in patients who did not undergo LDR, while patients in the LDR group did not experience grade II or above esophagitis reactions, and the difference was statistically significant (p = 0.039). Except for two cases of leukopenia in the group that did not undergo LDR (recovered after suspending radiotherapy and symptomatic supportive treatment), no grade IV side effects were observed in the remaining patients. The distribution of the main acute toxic side effects in the two groups patients is shown in Fig. 2. The correlation analysis between side effects and LDR in patients who have received previous radiotherapy is shown in Table 3. From a pathological perspective, both groups showed good response rates for Lumina A and Triple Negative types, but the difference was not statistically significant (LDR group: x2 = 4.791, p = 0.571; LDR group not treated: x2 = 2.919, p = 0.891). See Fig. 3.

Figure 2 Distribution of main acute side effects in two groups patients.

Table 3 Correlation analysis between side effects and LDR in patients who have received previous radiotherapy.

		Previous RT	
		LDR group	No LDR group	
Dermatitis	x 2	0.885	0.849	
	p	0.347	0.033	
Soft tissue injury	x 2	0.147	8.617	
	p	0.929	0.035	
Granulocytopenia	x 2	13.677	9.502	
	p	0.003	0.050	
Anemia	x 2	3.659	1.530	
	p	0.301	0.675	
Thrombocytopenia	x 2	0.529	0.707	
	p	0.409	0.815	
Swallowing discomfort	x 2	6.518	4.523	
	p	0.011	0.104	
Nausea	x 2	7.222	7.583	
	p	0.007	0.023	
Note:

LDR, low-dose radiation; RT, radiation therapy.

Figure 3 Distribution of recent efficacy analysis of two groups patients with different pathological types.

Comparison of recent therapeutic effects between two groups

Compared with the non LDR group, the CR (Complete Response) rates were 42.3% vs. 38.5%, p = 0.777, the PR (Partial Response) rates were 53.8% vs. 50.0%, p = 0.781, and the ORR (Objective relief rate) were 96.2% vs. 88.5%, p = 0.833, respectively.

There was one case of SD (Stable Disease) in the LDR group, and three cases of SD in the non LDR group, with a difference of 3.8% vs. 13.0%, p = 0.303. Neither group of patients developed PD (Progressive Disease). The specific recent therapeutic effects are shown in Table 4.

Table 4 Recent efficacy analysis of two groups patients.

	CR	PR	SD	PD	
LDR group (n)	11	14	1	0	
No LDR group (n)	10	13	3	0	
p	0.777	0.781	0.303		
Note:

LDR, low-dose radiation; CR, complete response; PR, partial response; SD, stable disease; PD, progressive disease.

Comparison of survival status between two groups

The 3-year OS and LFRS comparison between the two groups of patients was 54.8% vs. 55.8%, p = 0.845 and 54.2% vs. 52.9%, p = 0.751, respectively. The 3-year DMRS comparison between the two groups of patients was 67.9% vs. 51.9%, p = 0.097. The 3-year DSS comparison between the two groups of patients was 39.2% vs. 49.6%, p = 0.993. The OS, LRFS, DMFS and DSS survival curves of the two groups patients are shown in Fig. 4.

Figure 4 (A–D) The OS, LRFS, DMFS and DSS of two groups.

Discussion

Postoperative recurrence of breast cancer refers to the recurrence of the same nature after modified radical surgery, excluding the second primary cancer. A comprehensive evaluation is usually required for such patients, including tumor burden, pathological type, molecular typing, presence of distant metastases, systemic status, pathological type of recurrence, immunohistochemical results, and previous treatment plans (Ma et al., 2021).

Due to the temporal and spatial heterogeneity, the molecular classification of recurrent lesions may not be completely consistent with that of the primary lesion. Currently, it is believed that biopsy of recurrent lesion should be performed to determine the molecular classification (Demicheli et al., 2010). If the recurrent lesion cannot undergo R0 resection after multidisciplinary evaluation, radiotherapy should be considered, supplemented by a reasonable systemic treatment plan at the same time (Gradishar et al., 2022; Arthur et al., 2020; ElSherif et al., 2022; Meattini et al., 2024).

In order to reduce side reactions, LDR is gradually receiving attention (Kaidar-Person, Oldenborg & Poortmans, 2018; Walstra et al., 2021; Zielinski et al., 2009). In 1986, the United Nations Scientific Committee on the Effects of Atomic Radiation defined LDR as low energy transfer radiation with a dose within 0.2 Gy or high energy transfer radiation with a dose within 0.05 Gy, and radiation with a dose rate within 0.05 mGy/min (Zielinski et al., 2009). At present, in vitro and in vivo experimental studies suggest that it may have a protective effect on normal tissues through excitatory effects and adaptive responses (Weissmann et al., 2022; Wang et al., 2021; Khan et al., 2021). Pre-clinical data confirms that LDR reactions mostly occur between 40–80 cGy (Tang & Loke, 2015). Applying this mode to anti-inflammatory treatment in the body has shown efficacy (Niewald et al., 2022).

In our study, adding LDR before radiotherapy showed good tolerance and no significant increase in side effects. Compared to the group without LDR, the LDR group showed a certain degree of normal tissue protection, with less incidence of grade II or above side effects in the skin and soft tissue system, no grade II or above esophagitis reactions, and no grade IV leukopenia. The results are similar to a prospective controlled randomized clinical trial in phases II–III (Al-Rajhi et al., 2021). Al-Rajhi et al. (2021) included 108 eligible nasopharyngeal carcinoma patients of III–IVB in the study. The patients were randomly assigned to either the experimental group (54 patients) for induction chemotherapy with LDR (0.5 Gy, twice a day, 6 h apart, for 2 days) or the control group (54 patients) for induction chemotherapy alone. The results showed no significant difference in toxicity between the two treatment groups. Ngoi et al. (2021) explored the efficacy and safety of low-dose whole abdominal radiotherapy combined with paclitaxel in platinum resistant ovarian cancer. The results showed that the side effects were tolerable, and common grade ≥3 adverse events were neutropenia (60%) and anemia (30%). In addition, from Table 3, we found that LDR seems to have a better protective effect on normal tissue damage from re radiotherapy for patients who have previously received radiotherapy, especially in terms of skin and soft tissue. More data is needed to support this conclusion.

In this study, there were certain differences in the response rates of different pathological types compared to previous studies. In the neoadjuvant chemotherapy of locally advanced head and neck squamous cell carcinoma, LDR sensitization was combined with a total radiation dose of 640 cGy, and sensitization was performed for two cycles. The response rate was 82% (Gleason et al., 2013). The response rate of neoadjuvant chemotherapy combined with LDR sensitization in locally advanced breast cancer was significantly higher than that of non sensitized group (36 cases) (88.9% vs. 69.5%), (p < 0.05) (Nardone et al., 2014). In the other 20 cases of stage IIa–III a breast cancer, the pathological response rate was 33.3%, of which the general response rate of 2/11 Luminal A and 4/6 Luminal B was 35.3% (Bufi et al., 2012). Another study selected patients with recurrent and refractory NSCLC and chose the combination therapy of pemetrexed and LDR, with a total response rate of 42% (CR 2/19; PR 6/19) (Mantini et al., 2012). Considering the differences in pathological types and staging between studies, it is worth further stratified research with a larger sample size.

The group with LDR showed better efficiency than the group without LDR, but the difference was not statistically significant. Pre-irradiation with LDR did not show any worse 3-year OS, LRFS, or DSS compared to the group without LDR. Moreover, pre-irradiation with LDR seems to have a better DMFS (p = 0.097). Further confirmation is needed by conducting randomized controlled prospective clinical studies in the future. A phase II clinical study of LDR combined with neoadjuvant chemotherapy in locally advanced head and neck squamous cell carcinoma (Gleason et al., 2013) reported long-term follow-up results: with a median follow-up of 83 months, the local control rate was 80%, the distant control rate was 77%, the 5-year OS was 62%, the 5-year DSS was 66%. The 5-year follow-up showed a better prognosis than historical controls, similar to existing authoritative therapies. Research has confirmed that there is also a 30% response rate in pancreatic cancer (Morganti et al., 2014) and intestine cancer (Regine et al., 2007). It can be seen that adding LDR before treatment did not reduce long-term efficacy.

This is a retrospective study with a small sample size, which may lead to biased results. At the same time, the matching method uses a 1:1 propensity score matching (PSM) method for patients who receive treatment simultaneously. Using different variables to balance patient characteristics can also produce different results. Additionally, this experiment is a single center study. It is worth expanding the sample size, conducting prospective, multicenter clinical studies to further confirm.

In conclusion, compared with traditional radiotherapy, LDR pre irradiation has better tolerance and efficacy in patients with local chest wall recurrence of breast cancer after surgery. It is worth further exploring the possibility of this model becoming another comprehensive treatment plan for clinical treatment.

Supplemental Information

Supplemental Information 1 Data.

Supplemental Information 2 Codebook for categorical data.

Additional Information and Declarations

Competing Interests

Author Contributions

Human Ethics

Data Availability

The authors declare that they have no competing interests.

Ruifang Zeng performed the experiments, authored or reviewed drafts of the article, and approved the final draft.

Hanyu Wang performed the experiments, authored or reviewed drafts of the article, and approved the final draft.

Xiaojun Cai performed the experiments, authored or reviewed drafts of the article, and approved the final draft.

Junhao Lin analyzed the data, prepared figures and/or tables, and approved the final draft.

Pengfei Li analyzed the data, prepared figures and/or tables, and approved the final draft.

Shan Zeng conceived and designed the experiments, authored or reviewed drafts of the article, and approved the final draft.

Aimin Li conceived and designed the experiments, authored or reviewed drafts of the article, and approved the final draft.

The following information was supplied relating to ethical approvals (i.e., approving body and any reference numbers):

The Ethics Committee of the Integrated Hospital of Traditional Chinese Medicine, Southern Medical University approved the experimental and research protocol of this study (ID:201902SB-016-09). All procedures performed in this study were according with the ethical standards of the institutional research committee and with the 1964 Helsinki declaration and its later amendments or comparable ethical standards. Written informed consent was provided and signed by all patients prior to sample collection.

The following information was supplied regarding data availability:

The raw measurements are available in the Supplemental File.

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
