# Peer review of "Effect of low-dose radiation pre-irradiation on postoperative local chest wall recurrence of breast cancer—A retrospective study"

_PeerJ, doi:10.7717/peerj.18717_

## Round 0.1 · original submission · Minor Revisions

The introduction would be strengthened by including current evidence of LDR applications in other cancer types, particularly existing small-sample clinical studies in ovarian and head and neck cancers, and specifically addressing its potential application in recurrent breast cancer.

Regarding methodology, we recommend clarifying the tumor histopathology variables used in the Propensity Score Matching (PSM) process and standardizing the terminology between the methods section and Table 1. Additionally, the chemotherapy section would benefit from more detailed information about the specific regimens employed and their relative proportions in the study population, as different combinations may significantly impact outcomes, particularly regarding bone marrow suppression.

The results section requires some refinement in its presentation. The description of side effects should be revised to more accurately reflect that LDR group patients predominantly experienced Grade I effects rather than suggesting an increased incidence of side effects. All clinical response abbreviations (CR, PR, ORR, etc.) should be defined at first use for clarity.

Given the retrospective nature of the study, we suggest expanding the discussion to address potential methodological limitations, particularly regarding the PSM approach and its potential impact on results.

Reviewer 1 ·

Basic reporting

At present, small sample clinical studies have applied LDR to other types of cancer, such as ovarian cancer and head and neck cancer, and the results show that it has certain safety and efficacy. However, whether LDR is applied to recurrent breast cancer can be briefly described in the introduction section.

Experimental design

In the materials and methods section, lines 92-93, use the Propensity Score Matching (PSM) method to balance the patient characteristics. Specifically, what does the variable of tumor histopathology refer to? Table 1: The clinical characteristics of two group patients, only the differentiation variable. It should be standardized and explained.

Validity of the findings

This study investigated to determine the efficacy and safety of low-dose radiotherapy (LDR) for postoperative local chest wall recurrence of breast cancer, which has certain innovative significance.

Additional comments

This study investigated to determine the efficacy and safety of low-dose radiotherapy (LDR) for postoperative local chest wall recurrence of breast cancer, which has certain innovative significance. However, there are still several points in the article that need further revision:
1. At present, small sample clinical studies have applied LDR to other types of cancer, such as ovarian cancer and head and neck cancer, and the results show that it has certain safety and efficacy. However, whether LDR is applied to recurrent breast cancer can be briefly described in the introduction section.
2. In the materials and methods section, lines 92-93, use the Propensity Score Matching (PSM) method to balance the patient characteristics. Specifically, what does the variable of tumor histopathology refer to? Table 1: The clinical characteristics of two group patients, only the differentiation variable. It should be standardized and explained.
3. Line 165-167, compared to patients who did not receive LDR, more patients in the LDR group exhibited only grade I side effects in their skin and soft tissue systems, which did not reach grade II, rather than a higher incidence of grade I side effects. This result should be accurately described again, otherwise, it may lead to misunderstandings that adding LDR results in higher grade I side effects.

·

Basic reporting

no comment

Experimental design

1. In the materials and methods section, lines 118-120, the main chemotherapy drugs are listed here, but the combination of different drug regimens can have different effects on the research results. The main regimens should be listed or the proportion of the main regimens should be provided, because the use of two drug regimens compared to the use of three or more drugs has a significant difference in bone marrow suppression.
2. In the result section, lines 118-120, the full names of CR, PR, ORR, SD, and PD should be clearly stated here, not just abbreviations.

Validity of the findings

The study is a retrospective study, which may cause bias in the results. At the same time, the matching method uses a 1:1 propensity score matching (PSM) method for patients who receive treatment simultaneously. Using different variables to balance patient characteristics can also produce different results, which should be explained in the discussion section.

---

## Round 0.2 · accepted · Accept

All comments have been addressed by authors. And, I find all statistical analysis methods were performed correctly.

Therefore, I think this paper can be accepted for publication.

Reviewer 1 ·

Basic reporting

no comment

Experimental design

no comment

Validity of the findings

no comment

Additional comments

no comment